# Dairy Cattle Euthanasia—Focus Groups Exploring the Perspectives of Brazilians Working in the Dairy Cattle Industry

**DOI:** 10.3390/ani12040409

**Published:** 2022-02-09

**Authors:** Victoria R. Merenda, Eduardo B. de Oliveira, Heather N. Fowler, Monique D. Pairis-Garcia

**Affiliations:** 1Department of Population Health and Pathobiology, College of Veterinary Medicine, North Carolina State University, Raleigh, NC 27606, USA; vrmerend@ncsu.edu; 2Department of Population Health and Reproduction, College of Veterinary Medicine, UC Davis, Tulare, CA 93274, USA; ebarrosdeoliveira@ucdavis.edu; 3College of Veterinary Medicine, Iowa State University, Ames, IA 50011, USA; hnfowler@gmail.com

**Keywords:** animal welfare, focus groups, euthanasia, Brazilian, dairy cows

## Abstract

**Simple Summary:**

Euthanasia is the practice of ending the life of an animal that has no possibility of improvement and aims at minimizing suffering and mitigating poor animal welfare. Concerns regarding animal welfare are an international priority given that euthanasia standard requirements have the potential to impact the global trade of animal products. To ensure positive dairy cattle welfare by minimizing suffering via euthanasia, we must first understand how euthanasia is viewed within the Brazilian dairy community and identify barriers that prevent timely euthanasia from occurring. Therefore, we aimed to explore perspectives and attitudes about euthanasia specific to the Brazilian dairy industry using focus groups. Upon analysis of the discussions, three main themes were revealed: Euthanasia Training and Farm and Human Components. Several subthemes are discussed. The lack of nationally recognized euthanasia guidelines for dairy cattle paired with ineffective and inaccessible euthanasia tools makes it difficult for dairy veterinarians to implement humane protocols for on-farm euthanasia. In addition, logistical factors, particularly, the financial cost of euthanasia and the human–animal bond, play a role in the failure to perform euthanasia when warranted. Future studies should focus on the development of science-based standards and producer training to improve the consistency of on-farm euthanasia in Brazilian dairy operations.

**Abstract:**

The objective of this study was to explore perspectives and attitudes about euthanasia specific to the Brazilian dairy cattle industry. Twenty-five Brazilian citizens (13 veterinarians, 4 animal scientists, 3 professors, 3 researchers, 1 dairy owner, and 1 caretaker) participated in one of three focus groups conducted and recorded online (10, 8, and 7 participants per group). Questions regarding euthanasia were posed by a moderator, and the focus group discussions were then transcribed verbatim for analysis. After the initial data analysis, themes were evaluated and collapsed into three major categories: Euthanasia Training and Farm and Human Components. A complex interconnection between the three main themes and multiple subthemes specific to dairy cattle euthanasia was also revealed. The lack of nationally recognized euthanasia guidelines for dairy cattle paired with ineffective and inaccessible euthanasia tools makes it difficult for dairy veterinarians to implement humane protocols for on-farm euthanasia. In addition, logistical factors, particularly, the financial cost of euthanasia and the human–animal bond, play a role in the failure to perform euthanasia when warranted. Future studies should focus on the development of science-based standards and producer training to improve the consistency of on-farm euthanasia in Brazilian dairy operations.

## 1. Introduction

Euthanasia is the practice of ending the life of a patient or animal who has no prospect of improvement [1]. Euthanasia minimizes suffering and pain experienced by the individual and is an ethical responsibility for caretakers designated to this role [2]. Thus, animal caretakers must have not only the proper training to identify compromised animals, but also the decision-making skills and confidence to perform euthanasia when necessary. Nonetheless, even with extensive euthanasia training and experience, individuals performing this task can be affected emotionally. Euthanasia-related stress has been documented in those working in shelters [3], veterinary clinics [4], swine operations [5,6,7], and dairy farms [8,9,10]. This condition, more commonly referred to as compassion fatigue, affects people in caretaking jobs and may interfere with an individual’s ability to perform work-related tasks [11].

In recent years, animal welfare scientists have begun to explore perceptions and attitudes towards euthanasia with a particular focus on emotional barriers associated with performing this act. This work, specific to dairy cattle operations, has highlighted perspectives from both dairy farm caretakers [8,9] and veterinarians [10] and has shown that although most felt comfortable performing euthanasia, compassion fatigue and emotional stress related to making the decision were evident.

This issue is not unique to the United States (U.S.A.), and concerns regarding animal welfare and suffering are an international priority, given euthanasia standard requirements have the potential to impact the global trade of animal products [12,13].

Brazil has the third largest dairy herd in the world, with 16.4 million dairy cows [14] and an annual milk production of 24.9 million metric tons [15]. The Brazilian dairy industry is composed primarily of smaller farms with over 200,000 producers responsible for 82% of national milk production [16]. Given this, production practices and management strategies vary widely [15] and to date, there are no national standards outlining best management practices, such as timely euthanasia decision making for dairy herds.

Given the structure of the Brazilian dairy industry and the lack of specific management standards and guidelines, ensuring humane and timely euthanasia of compromised cattle is critical. This pertains not only to individual cows requiring euthanasia on farms but also to zoonotic outbreak events in which large populations of cattle require euthanasia [17,18].

In order to ensure positive dairy cattle welfare by minimizing suffering via euthanasia, we must first understand how euthanasia is viewed within the Brazilian dairy community and identify barriers that prevent timely euthanasia from occurring when needed. Therefore, the objective of this study was to explore perspectives and attitudes about euthanasia specific to the Brazilian dairy cattle industry using focus groups.

## 2. Materials and Methods

All research was reviewed and approved by the North Carolina State University IRB Committee for Human Subjects Research (protocol #19243).

### 2.1. Participant Recruitment

Brazilian citizens associated with the dairy cattle industry (i.e., veterinarians, academia, producers, caretakers) were recruited using an electronic mailing list to participate in this study using a convenience sampling methodology. The electronic mailing list was initially created by an academic professor who invited Brazilian colleagues with dairy cattle experience to join. This mailing list has been used as a form of communication for this group, including, but not limited to, research collaboration, projects of interest, and research advice. Specifically for this study, a private message was sent to each individual in the group, and a series of questions were asked to determine if the group’s members were (1) familiar with dairy cattle; (2) familiar with euthanasia in dairy cattle; (3) willing to participate in a focus group discussion. In order to participate, individuals were required to be Brazilian citizens and native Portuguese speakers and to have experience with dairy cattle euthanasia. Following recruitment, 27 individuals agreed to participate in one of three focus groups. The distribution of the individuals in the focus groups was according to the participants’ schedule availability.

### 2.2. Web-Based Focus Group

Focus groups were conducted using the online platform Zoom (Zoom Video Communications Inc., San Jose, CA, USA, 2016). Two weeks prior to the focus group, each participant received an individual email to access the electronic consent form (Qualtrics International Inc., Seattle, WA, USA), a demographic survey and the zoom link. A reminder text message was sent the day prior to the focus group to ensure participants arrived on time. All participants were made aware of the fact that the focus group would be recorded and were reminded that participation was voluntary and that they could remove themselves from the zoom meeting at any time. An incentive ($25 gift card) was provided to all participants who completed the study.

Focus groups were conducted in Portuguese and audio-recorded to support future transcription (total duration per group, 90 min). One moderator and one assistant were present in all three focus groups to coordinate the discussions (both native Portuguese speakers). The moderator was a PhD student with experience in the dairy industry and expertise in cattle euthanasia, and the assistant was a DVM resident with an expertise in cattle health. The moderator and assistant also served as coders for the analysis of this study.

Discussion was prompted using the same eight questions previously utilized in work conducted by Wagner and colleagues [9,10]. All questions were translated into Portuguese and included: (1) What comes to mind when you think about euthanizing animals on-farm? (2) What, if any, are the benefits of euthanizing animals on-farm? (3) What, if any, are the drawbacks to euthanizing animals on-farm? (4) When do you know it is the right time to euthanize an animal? (5) When do you know it is not the right time to euthanize an animal? (6) What are the main reasons why you would delay euthanasia? (7) What are the main reasons why you would not perform euthanasia? (8) What other factors might you consider when making the decision to euthanize animals on-farm? Follow-up questions were asked when necessary by the moderator to further encourage discussion.

Focus groups were transcribed and analyzed by two independent coders (moderator and assistant) using the thematic analysis described by Braun and Clarke [19]. The overall agreement between the coders was 94% before consensus was reached. The misinterpretation of the codes was discussed between coders until final agreement was reached.

## 3. Results and Discussion

Twenty-five Brazilian citizens associated with the dairy cattle industry participated in one of three focus groups (10, 8, and 7 participants/group, respectively). Two participants originally recruited did not participate due to undisclosed personal reasons. The majority of the participants identified themselves as veterinarians (52%), followed by animal scientists (16%), professors (12%), researchers (12%), dairy owners (4%), or caretakers but not owners (4%). Eighty-four percent of the participants responded “yes” when asked if any cow had been euthanized during the last 12 months where they worked, reflecting that the majority of the individuals participating in the focus groups were familiar with euthanasia on-farm.

### 3.1. Thematic Analysis

Initial analysis of the data revealed a complex interconnection of multiple themes and subthemes specific to dairy cattle euthanasia (Figure 1). After the initial data analysis, themes were evaluated and collapsed into three major categories: Euthanasia Training and Farm and Human Components.

### 3.2. Theme 1: Euthanasia Training

This theme included any discussion revealing euthanasia misconceptions, lack of euthanasia equipment, and science-based guidelines to support the development of euthanasia protocols.

Based on the focus groups, the term ‘euthanasia’ is not used frequently in the Brazilian dairy industry and was consistently incorrectly defined as a procedure that relies exclusively on the use of anesthetics. One of the veterinarians stated:


*“We commonly do not use the term euthanasia in milk production. We say culling. When we talk about registering in an online system (…), we write culling, but it's actually death… Euthanasia is a term that not even the producers know very well what it is (…) So when we say that the cow was culled, it does not always mean that it went to the slaughterhouse, death is considered culling. Euthanasia is unknown to producers. For veterinarians… Look, I haven't done a ‘properly said’ euthanasia since the residency period. (…) In Brazil, it is practically impossible to perform euthanasia like we used to do in the hospital.”*


According to work conducted by Figueiredo and Araujo [20], euthanasia is a controversial subject that is poorly addressed in Brazilian veterinary curriculum. Given Brazil has no national standards specific to cattle euthanasia, many veterinarians rely on external resources to develop euthanasia protocols, including the American Veterinary Medical Association euthanasia guidelines (AVMA; Schaumburg, IL, USA). Although a comprehensive document, these guidelines were developed with a U.S.-specific focus. For example, AVMA guidelines identify penetrating captive bolt guns, firearms, and barbiturate overdose as primary mechanisms for euthanasia in cattle. However, in Brazil, captive bolts gun use on-farm is minimal [20], firearm ownership is extremely restricted and must be obtained through a federal police registration database (regardless of the recent law approval for easing gun ownership) [21,22], and legal frameworks safeguarding pharmacovigilance in the country limit barbiturate access and use [23].

It was also identified that the tools needed to perform euthanasia were not always available. One participant expressed frustration over this fact stating


*“I use xylazine with potassium chloride. I don’t know if this is the right protocol, but this is what I use here.”*


Another participant complemented the discussion and expressed concern about the accessibility of the drugs


*“The potassium chloride is not easy to find, it is not easy to buy. The situation is very complicated.”*


Deficiencies in euthanasia guidelines and equipment specific to the Brazilian dairy industry have limited Brazilian veterinarians in developing and implementing euthanasia protocols that can be used on-farm. As nicely summarized by one participant:


*“The disadvantage is that we do not have protocols. Because we don’t have protocols, we end up doing a handful of things that we do not agree with, (…) We must be realistic about things. Anyone who works in the field knows. We use (…) products that end up causing animal suffering…”*


Future work is needed to develop science-based guidelines specific to the Brazilian dairy industry that can support the development of realistic protocols as well as the identification of effective and humane euthanasia methods. Previous work conducted by Dalla Costa and colleagues [24] may be used as a template to move the Brazilian dairy industry forward. Dalla Costa and colleagues [24] evaluated on-farm euthanasia methods and attitudes towards euthanasia in the Brazilian swine industry using survey methodologies. This work identified current euthanasia methods used on pig farms and identified potential barriers for implementing euthanasia successfully. This work was later used to develop national guidelines for swine euthanasia and identified key conditions in swine, which warrant immediate euthanasia [25]. Given the challenges that Brazilian swine and dairy producers face are similar, collecting information on dairy farm euthanasia may assist in identifying knowledge gaps that can be filled through education and research and in developing national standards on cattle euthanasia.

### 3.3. Theme 2: Farm Components

The second theme was centered around logistical factors impacting euthanasia, including cost (labor, time, equipment) and farm size. Logistical concerns specific to the financial impact were consistently brought up in each focus group. Balancing the cost–benefit relationship when treating a cow was evident, with many participants acknowledging that dairy producers do not have the financial resources to support long-term or aggressive treatments. For example, one participant stated:


*“Maybe you would have a solution in the case of a fracture, for example, but in a production system this [the treatment] is not economically viable.”*


Not only does treating a cow have significant financial impact on the operation, but also many participants posed concerns specific to the success of such treatment interventions. As mentioned by one participant:


*“I prefer my employee to check on 50 calves per hour preventing something than losing one hour treating an animal that will not bring any result.”*


This suggests that either the treatment protocols available to Brazilian dairy producers are not effective or the condition of the animal is beyond recovery and any treatment is unlikely to be successful.

This issue is not unique to the Brazilian dairy industry and has received much attention in the last 5 years in the U.S.A. [26]. A good case example of this is the management of non-ambulatory cows on U.S. dairy farms. It is estimated that approximately 19% of cows not standing for at least 24 h are involuntarily culled and sold for slaughter in the U.S.A. [27], and it can be assumed that this number may be even higher if counting non-ambulatory cattle that die on-farm. Work assessing treatment interventions suggests that cows that are non-ambulatory for more than 24 h are unlikely to recover, particularly when the cause is not associated with hypocalcemia [28]. Even with this information, 46.3–54.9% of the surveyed U.S. producers would treat and monitor non-ambulatory cattle [9] regardless of the high likelihood for poor welfare outcomes for these individual cows. The sentiment to administer and treat cattle that are unlikely to recover was also shared by one Brazilian citizen in the focus group stating:


*“You either treat, treat, treat and it [cow] dies or you treat it and she doesn’t die.”*


Providing treatment to cattle with a poor likelihood of recovery has both negative welfare and economic implications to the industry. Ensuring appropriate training of veterinarians and effective communication to producers is imperative when treatment decisions are being made. Animals whose conditions are refractory in nature must be euthanized, given that treatment will not change the end outcome for the cow, and failure to provide any intervention will result in prolonged suffering and duress [2].

Euthanasia implementation was also directly influenced by farm size. The Brazilian dairy industry is diverse and composed of large (>101 cows, 10% of farms), medium (11–100 cows; 57% of farms), and small operations (1–10 cows; 33% of farms) [15,29]. Based upon responses from focus group participants, smaller farms tended to demonstrate greater challenges when implementing euthanasia consistently. For instance, one participant said:


*“The smaller the farm, the larger is the amount of time this animal will be treated. They do not want to lose this animal because this animal has a huge value for him [farmer].”*


And another stated:


*“When the animal is sick, it [euthanasia] depends on the size of the property, on the financial condition of the owner, on the attachment to the animal (whether the owner is attached to the animal or not), and if he has time and employees for that.”*


The USDA NAHMS [26] reported a similar phenomenon in the U.S. dairy industry, highlighting that a higher percentage of non-ambulatory cows died unassisted on smaller farms (30 to 99 cows; 23.6%) than on operations with more than 500 cows (14.8%). Furthermore, larger farms seem to make euthanasia decisions in shorter time periods than smaller farms [26], and this suggests that external factors are likely to play a larger role in the euthanasia decision-making process than the cow condition alone. Given these variations, future educational programming must take into account farm demographics and ensure that all producers, regardless of their farm size, can implement euthanasia effectively and consistently.

### 3.4. Theme 3: Human Components

The last theme of the focus groups included the impact of the participants’ emotional response on the decision-making process and public perception on conducting euthanasia.

Euthanasia on livestock operations is not easy to discuss, and euthanasia decisions are not easy to make. Making the decision to euthanize depends on a multitude of factors including the emotional response and acceptance of the act. When participants were asked the question *“When is the right time for euthanasia?”* the majority of the responses were not based on clinical signs or conditions of the animal, but on how the person performing euthanasia perceived the act, as one participant stated:


*“I think… the right time for euthanasia has more to do with the people than with the animal condition”.*


Another individual stated:


*“I choose euthanasia when I am confident that this was the best decision I could have made for that animal and that I will lay my head on the pillow and think ‘I’ve tried everything I could.”*


The emotional impact on the decision to euthanize may be impacted by religious affiliation and human–animal bond. In Brazil, it is estimated that 93% of the citizens have a religious affiliation, with the majority being Christian [30]. In addition, the majority of citizens entering the field of veterinary medicine (i.e., veterinary students) believe that animals have a soul [31], over 90% of Brazilian dairy farmers surveyed in Southern Brazil could recognize all their individual dairy cows, and over half of those cows had a name [32]. As defined by the AVMA [33], the human–animal bond is “a mutually beneficial and dynamic relationship between people and animals that is influenced by behaviors essential to the health and wellbeing of both. This includes, among other things, emotional, psychological, and physical interactions between people, animals, and the environment.” The strong bonds that dairy producers and caretakers develop with their cattle may result in poor welfare outcomes for these cattle if that relationship prevents an individual from making timely euthanasia decisions [26]. In situations such as these, those individuals that have developed strong bonds may not be able to serve as an objective proxy for the animal, which results in prolonged treatment and an unassisted death [2]. Future work is needed to better understand the complex and dynamic relationships between farm animals and their caretakers.

Confidence around making and performing euthanasia varied greatly within these focus groups. Although some participants felt confident with making the decision, others, such as one dairy owner, responded:


*“I always delay euthanasia. There is a calf here that it has been a month that we’ve been trying to fix her lesion and she doesn’t get better. But at some point, you reach the conclusion that there is no other way.”*


Ensuring producers and veterinarians are confident in the decision-making process is key to guaranteeing timely euthanasia is implemented consistently on farm. Work conducted by Campler and colleagues [34] demonstrated that trained and confident swine caretakers were more comfortable conducting complex decision making on euthanasia and performing euthanasia than caretakers who did not feel confident. Even though individuals participating in the focus had extensive experience within the dairy industry, gaining confidence through educational training programs would likely result in more consistent and reliable euthanasia decision making.

In addition to the emotional response and concerns related to performing euthanasia by those directly involved with the dairy industry, many participants noted the effect of public perception on performing dairy cattle euthanasia. More recently, the Brazilian dairy industry has received much scrutiny regarding animal management, specifically, regarding the euthanasia of healthy bull calves on-farm. In work conducted by Cardoso and colleagues [35], 79% of the surveyed participants were unaware that the Brazilian dairy industry “kills” bull calves immediately after birth, and 90% of those participants rejected the practice as acceptable [35].

Bull calf management was frequently mentioned in the focus groups, with one participant noting


*“Every week, there is a new video on Instagram about male calves in the dairy production. We need to solve this problem.”*


Focus group participants felt that this scrutiny has limited their ability to perform euthanasia, as stated


*“When you have an audience that you don’t know a lot, you may have to wait a little, change the environment, think of your surroundings before [performing euthanasia].”*


Euthanizing healthy calves is not just a concern for the general public [2,36] but also a contentious issue amongst those participating in the focus group. Participants in the focus group both supported and condemned the practice, while a few individuals removed themselves from the conversation. One participant in particular confessed that she would leave the farm whenever bull calf euthanasia was performed to avoid moral distress.

As the public in Brazil becomes further removed from livestock practices, the Brazilian dairy industry must address how euthanasia is perceived and accepted in the country, as well as how management practices must change and evolve to fulfill public demands specific to dairy cattle welfare.

## 4. Conclusions

In our assessment of perspectives surrounding euthanasia in the Brazilian dairy industry, our focus groups revealed euthanasia training and farm and human components as primary factors influencing the euthanasia decision-making process on a farm. The lack of nationally recognized euthanasia guidelines for dairy cattle coupled with ineffective and inaccessible euthanasia tools makes it difficult for dairy veterinarians to implement humane protocols for on-farm euthanasia. Lastly, logistical factors, particularly, the financial cost of euthanasia and the human–animal bond, play a role in the failure to perform euthanasia when warranted. Future research must focus on the development of science-based standards and producer training to improve the consistency of on-farm euthanasia for Brazilian dairy operations.

## Figures and Tables

**Figure 1 animals-12-00409-f001:**
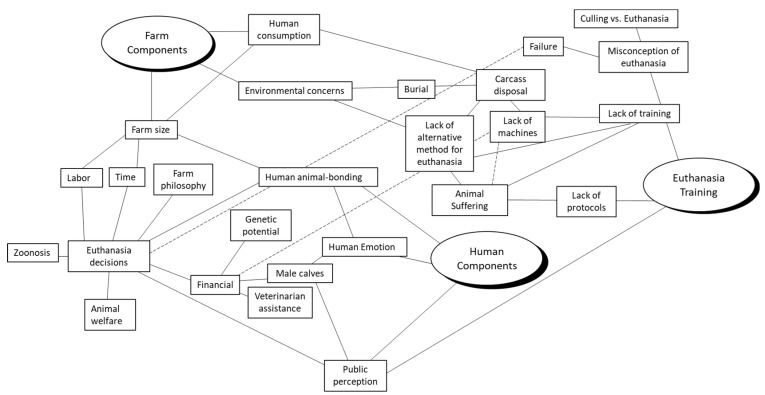
Initial thematic map.

## Data Availability

The data presented in this study are available on request from the corresponding author. The data are not publicly available due to ethical reasons.

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
