# Peer review of "Dairy Cattle Euthanasia—Focus Groups Exploring the Perspectives of Brazilians Working in the Dairy Cattle Industry"

_animals, 2022, doi:10.3390/ani12040409_

Round 1

Reviewer 1 Report

Dear authors,

You have written an interesting paper about a very important topic, and as a person without prior knowledge of Brazilian dairy farming I learned new things.

I have a few suggestions that might improve your work:

L85: How did you obtain the mailing list and who created it to begin with?

LL122-130: How were the participants distributed among the three groups? What is the difference between an "animal scientist" and a "researcher". Are your focus groups representative for people working in the dairy industry, e.g. should you not have tried to include more dairy owners?

LL138ff: As a reader without prior knowledge of the situation in Brazil I am left a bit hanging here - if you don't use captive bolt, firearms or barbiturates, what do you then do? Could you please clarify/give examples of the methods and tools that are used for euthanasia. And who usually performs the euthanasia - the vet or the farmer?

General comment to the three themes: You could consider including subtitles to make reading easier.

Reviewer 2 Report

Well done on this Important work. A larger sample size would have improved this study. Please check for errors in punctuation and use of quotation marks.

Reviewer 3 Report

The current study is on a topic of relevance and general interest to the readers of Animal journal. The study attempts to explore perspectives and attitudes of Brazilian citizens working in the dairy cattle industry on dairy cattle euthanasia using qualitative research methods. I found the paper to be overall well written and much of it to be well described. Below I have some comments for the Authors to address:

I do not think that the title reflects well the scope of the paper. Reference to citizens is misleading as there were only professionals involved in the FGI (i.e. veterinarians, academia, producers, caretakers) and the study is qualitative in nature.

In the paper based on qualitative FGI data one would expect more information on it how the participants responded to each of the major themes posed by moderator. I think some aspects covered by FGI are not tackled - particularly related to: what, if any, are the benefits of euthanizing animals on-farm and what, if any, are the drawbacks to euthanizing animals on-farm?

What I miss in your paper is a reflection data analysis and result of the initial data analysis as presented in Figure 1.

In the conclusion you mention “our assessment of attitudes and perspectives surrounding euthanasia in the Brazilian dairy industry, our focus groups revealed euthanasia training, farm and human components as primary factors influencing the euthanasia decision-making process on the farm”. I miss in the conclusion more reflection on attitudes to euthanasia as promised in the title.

Please add quotation in separate paragraphs and probably using other font size for the convenience of the readers.

good luck!

Round 2

Reviewer 3 Report

Dear Authors,

I think the manuscript has been satisfactorily improved and can be processed further. 

good luck!